# Factors regarding the dog owner's household situation, antisocial behaviours, animal views and animal treatment in a population of dogs confiscated after biting humans and/ or other animals

Ineke R. van Herwijnen⬥*◐, Joanne A. M. van der Borg⬥‡, Chantal M. Kapteijn‡, Saskia S. Arndt, Claudia M. Vinke◐

Division of Animals in Science and Society, Faculty of Veterinary Medicine, Department of Population Health Sciences, Utrecht University, Utrecht, The Netherlands

◐ These authors contributed equally to this work.
‡ JAMB and CMK also contributed equally to this work.
* i.r.vanherwijnen@uu.nl

## Abstract

To examine the dog ownership factors characteristic to a population of dogs confiscated after a human and/ or animal-directed biting incident, we compared bite risk assessment reports of 159 confiscated dogs in the time frame 2008, 2009, 2010 (tf1) and of 215 confiscated dogs in the time frame 2020, 2021, 2022 (until mid-May; tf2). The reports were compiled by the same institute in a standardized format. We studied frequencies and chi-square pairwise comparisons (P<0.05) for 30 identified ownership factors. Overall (tf1 and tf2), 1,308 ownership factors were mentioned in the reports and reports mentioning ≥5 factors were twice as frequent in tf2 (38%) than tf1 (16%). Our data suggest that nine factors may in particular serve as a warning signal for biting incidents, as these factors were most frequently (≥15%) prevalent in the total of reported cases: having a multiple dog household, a dog reportedly roaming a neighbourhood without an owner, a dog's care tasks being transferred, a short leash and muzzle obligation served to the owner for a dog, an isolated and/ or confined keeping of a dog, a dog owner's (suspected) substance abuse, a dog owner's (suspected) animal abuse, a dog owner aggressing at confiscation of the dog and a dog owner being reported on for antisocial behaviours such as intimidation. Particularly, a dog owner's aggressive or antisocial behaviours and previous obligations to muzzle and short leash a dog (in our dataset often inappropriately adhered to by owners), may indicate that a proportion of owners of confiscated dogs, may not always be willing and/ or capable to guarantee societal safety. The results show that identification of dog ownership factors, might be useful for establishing biting incident policies and further studies should be done on preventive measures and controls.

**Data Availability Statement:** All relevant data are within the paper and its Supporting Information files.

**Funding:** The author(s) received no specific funding for this work.

**Competing interests:** The authors have declared that no competing interests exist.

## Introduction

Dogs and humans have lived together for a long time, with suggestions made for dogs to be amongst the first species to be domesticated by humans [1–3], although the route towards domestication and the exact start of the process are being debated still [4–6]. The relationship between dogs and humans has received much scientific interest [7]. It is therefore surprising that little scientific information seems to be available on a darker side of dogs and humans living together: that of those situations where dogs are confiscated by a governmental body due to biting a human and/ or animal.

A dog's biting can be highly problematic to victims, as well as the dog, the owner, the owner's household and society. For The Netherlands, data is lacking and not centrally registered for animal-directed dog bites [8, 9]. For human-directed dog bites, a study from 2007 and 2008 estimated a total of 136,000 yearly biting incidents [10]. The majority of these biting incidents required no (professional) treatment, an estimated 40,000 required treatment in family practices, 11,000 required treatment in emergency departments/ polyclinics and an estimated 300 victims needed to be admitted to hospitals on a yearly base [10]. Specifically, for biting incidents causing injuries that required hospital emergency department treatment due to open wounds and sometimes fractures, a more recent non-scientific study extrapolated data from fourteen emergency departments to total of 1,800 yearly victims (uncertainty margin: 1,100–2,600), of which an estimated 200 patients needed to be admitted to hospital [11]. In addition to physical damage, biting incidents can cause mental trauma to those involved [12–16] and costs for society, such as through medical costs [17–21]. For the dogs, consequences of biting may be relinquishment or euthanasia [22–25]. The consequences of a dog's biting, therefore, make it relevant to ensure that dogs' hereditary bases and their socialisation and habituation minimise the chances of biting [26–29]. In addition, we need to know specifically which ownership factors contribute to a dog causing a biting incident, as to predict which ownership situations need monitoring or attention in dog bite prevention policies and campaigns–strengthening these as to increase societal safety and safeguard animal welfare.

In this study we aim to add to scientific data specifically on biting incidents that are so severe that they lead to the confiscation of the biting dog by a governmental body. For these dogs, bite risk assessment reports were compiled by the same institute in a standardized format and suitable for systematically scrutinizing these for dog ownership factors. Availability of these reports over time, allowed us to detect possible trends of dog ownership factors for dogs confiscated after biting in the Netherlands and thus to contribute to filling a scientific gap. Presently, relatively few scientific studies cover the most severe cases of biting incidents or those study populations less likely to engage in research, but with a high urgency for prevention policies. We addressed the research question: 'Which dog ownership factors characterize a population of dogs confiscated after a biting incident with a human and/ or animal, in a more recent and a more historic time frame?'

## Methods

We analysed bite risk assessment reports made for the Dutch national and local government bodies for dogs confiscated after one or more biting incidents. The biting incidents could be with either human(s) and/ or other animal(s), including dog(s) and were studied for the time frames of 2008–2010 (tf1) and 2020–2022 (tf2) to assess dog ownership factors associated with this specific type of biting incidents. Additionally, we compared these earliest available and most recent time frames for possible changes over time.

## Bite risk assessments reports and the selection thereof

For our study on dog ownership factors in a population of dogs confiscated after a biting incident with human(s) and/ or animal(s) -including dogs-, we studied bite risk assessment reports on dogs confiscated by Dutch governmental bodies. We studied all reports made by Utrecht University for these governmental bodies, which regarded mostly the national governmental body of the Netherlands Public Prosecution Service (93%, N = 348) and in addition a smaller number of reports by municipalities (7%, N = 26). We chose two time frames. The first time frame (2008–2010, tf1) covered the start of bite risk assessment reports becoming a part of the process of assessing biting incidents, after a change in national legislation. The second time frame regarded the most recent frame of a comparable duration of years (2020-mid-May 2022, tf2).

A total of 374 bite risk assessment reports were included on dogs confiscated after one or more biting incidents. These reports were made by trained academic behavioural experts employed at Utrecht University following a standardized format. The reports regarded anonymous data only. The cases described in the reports regarded confiscated dogs as described above. These confiscated dogs were housed under regular animal shelter conditions after confiscation. The commission for an assessment was made through the Netherlands Enterprise Agency, by the Netherlands Public Prosecution Service as part of the confiscation procedure after biting incidents, in the majority of cases (N = 348). Similarly, a municipality could request a bite risk assessment report when a dog was confiscated by a local authority after a biting incident. This regarded the minority of cases (N = 26). Utrecht University assessed the risk of biting of the confiscated dog in a standardized way, based on four information sources, after receiving its assignment for a particular case. The first of these sources was the confidential and anonymized police or municipality report on the biting incidents. Each report was studied for information on the biting incident and biting damage, including any antecedent circumstances, potential stimuli for the dog to commence the biting and factors known on owner household and owner behaviours at the time of the biting incident and information on the dog's husbandry, keeping, care and owner-dog interactions. The second source was a veterinary report made within the first days of a dog's confiscation as to establish any medical conditions that could have attributed to the dog's biting behaviour (with added data on health status, such as neutering status, or from for instance X-rays if advanced diagnosis was deemed necessary, based on veterinary advice). These first two sources were relevant and used for our present study.

The third source was a standardized report on the dog's behaviour filled out by the dog's primary care taker during the sheltering after confiscation. This report was made after a minimal acclimatization time of two weeks, allowing the dog to settle in its new environment and form a social bond with one or more caretakers at the shelter. The fourth source regarded a behavioural observation. As a standard procedure, bite risk assessment reports were sent to the commissioner after finalisation and presented to a judge or the legal department of a municipality as an advice for further decision taking on the biting incident and measures to be taken, also with regard to minimizing future biting risk of the dog involved in a biting incident.

## Reports and data collection

For the present study we used the already drawn up and anonymous bite risk assessment reports and of these only the first two aforementioned sources of the police or municipality report and the veterinary report on the dog. The reports were studied for a total of eight owner household factors, seven owner antisocial behaviour factors, four husbandry/ animal view factors and eleven animal treatment factors which we determined based on an analysis of ten

randomly chosen bite risk assessment reports for factors mentioned in these reports. The list resulting from this first analysis was presented to three trained academic behavioural experts which had been involved in compiling bite risk assessment reports for at least two years. These employees checked the list for omissions. The resulting list of thirty factors was used for analysing a further ten bite risk assessment reports -again randomly chosen, but excluding the first ten reports- to check for any omissions, that is missed possible dog owner and dog ownership factors. The final list with thirty factors and descriptions is presented in four categories, in Table 1.

From each bite risk assessment report the information per factor was transferred to an Excel file and sorted per year of entry by one and the same researcher. After entering all data

**Table 1. Dog owner and dog ownership factors studied.** Dog owner and dog ownership factors studied, categorised into four categories: 'owner household', 'owner antisocial behaviours', 'husbandry/ animal view' and 'animal treatment'.

| |
|---|
| ***Owner household—Factors regarding household*** |
| Children placed in care/ court custody, child abuse or neglect |
| Criminal offences or subject known by police |
| Domestic violence |
| Financial issues |
| Form of assistance provided to owner |
| Living situation inadequate: dirty/ littered/ cluttered premises, disused premises, squatting, uninhabitable or maintenance neglect |
| Mental illness |
| Substance abuse |
| ***Owner antisocial behaviours—Factors regarding nuisance/ incidents with or without dog*** |
| Aggression towards police, bystanders, etc at confiscation of dog |
| Antisocial behaviours directed at humans, such as shouting, name calling, intimidation without dog (being mentioned) |
| Dog used as weapon or intimidation/ defence |
| Dog roaming without owner (unrelated to current biting incident) |
| Noise disturbance or other disturbance in neighbourhood (without dog) |
| Mentioning of previous dog(s) incidents or confiscation of previous dog(s) |
| Short leash/ muzzle obligation for this or previous dog |
| ***Husbandry/ animal view—Factors regarding dog and husbandry situation and animal views*** |
| Care task transfer |
| Illogical explanation of biting behaviour such as it being 'play' |
| Multiple dogs in household |
| Obtainment of dog |
| ***Animal treatment—Factors regarding animal treatment*** |
| Body Condition Score (BCS) too low |
| Isolated and/ or confined space |
| No access to (clean) water/ food |
| No access to daylight or covered bench |
| Old scars/ white hair markings |
| Parasites (endo-/ ectoparasites) |
| Physical punishment or corrective measure directed at dog |
| Skin/ coat/ nail issues (excluding nails damaged by finding grip during biting incident) |
| Suspected or established abuse |
| Unclean environment |
| Untreated pain and/ or illness signs |

into an Excel file both Excel and IBM SPSS Statistics 27 software were used for extracting descriptive data and for statistical analyses.

We present also on the severeness of dog bites, indicated in the reports as based on Ian Dunbar's biting scale [30]. In the reports, bites could either be scored as a clear 1, 2, 3, 4, 5, 6 or as fitting either one or another score, for instance as '3 or 4'. Therefore, we grouped scores into the categories of 'moderate', 'severe' or 'extremely severe' biting. The moderate category grouped scores of 1, 2, 3: up to four teeth punctures, none deeper than half the length of the dog's teeth. The severe category grouped '3 or 4', as well as '3 or 4 or 5': at least one puncture deeper than half the length of the dog's teeth, but not multiple bites. The extremely severe category grouped: '4 or 5', '4 or 5 or 6', as well as 5, 6: multiple bites with at least one puncture deeper than half the length of the dog's teeth and more severe, including death.

### Statistical analyses

For the 30 dog ownership factors we tested if these were present at different frequencies in tf1 and tf2, based on the total sum of factors (so unrelated to the cases). Next, we tested for different frequencies in tf1 and tf2 related to the 374 cases, including tests for multifactorial dog ownership situations. Multifactorial aspects of the cases were assessed as following. When studying the factors uncategorised, the cases with ≥5 reported factors were regarded as multifactorial. In addition, we studied if cases were multifactorial *within* each ownership category of 'owner household', 'owner antisocial behaviours', 'husbandry/ animal view' and 'animal treatment'. Here we regarded as multifactorial, those cases where ≥2 factors applied within a category, to a case. Finally, we determined for each of the 30 factors separately, how often each *individual* factor was reported on for each time frame.

We tested for differences between the two time frames of 2008–2010 and 2020–2022 (mid-May) with Pearson's Chi-square tests, regarding P-values of <0.5 as statistically significant and we indicate where Chi-square standardized residuals identify significant deviations from expected values (i.e. ≥|2|), in bold.

### Ethical considerations

For this study we used already available, yet anonymized data, as drawn up in the bite risk assessment reports of Utrecht University. Thus, we did not have to burden confiscated dogs, involved victims, dog owners, governmental bodies or any other involved parties, also avoiding unnecessary human psychological burdening. The study was approved by the Ethics Committee of the Faculty of Social and Behavioural Sciences of Utrecht University on June 30th 2022. As we used already available, fully anonymized data, no participants were involved and hence participant consent did not apply to our desk research study type.

## Results

### Owner characteristics

Owners were mostly (61%, N = 228 overall) registered as male (no information available on gender identity), with no significant difference between tf1 and tf2 for percentages of males, females, as detailed in Table 2.

### Dog characteristics

Dogs were between 8 months and 10 years old (median: 4, mode: 2; no significant difference between tf1 and tf2). Of all dogs included 66% (N = 245) was male, 34% (N = 128) was female (N = 1 missing value). Of all male dogs included 73% (N = 179) was intact, 13.5% (N = 33) was

**Table 2. Descriptive data on gender of owners of confiscated dogs.** Owner gender as count and as % of column total for a first time frame (2008–2010), a second time frame (2020-mid-May 2022) and overall for both time frames, with no identified significant difference between the two time frames (P≥0.05), for owners of 374 confiscated dogs.

| Owner gender | 2008–2010 | 2020–2022 | Both time frames |
|---|---|---|---|
| | N (% of column total) | N (% of column total) | N (% of column total) |
| Male | 99 (62%) | 129 (60%) | 228 (61%) |
| Female | 45 (28%) | 59 (28%) | 104 (28%) |
| Two owners | 8 (5%) | 20 (9%) | 28 (7%) |
| Unknown | 7 (5%) | 7 (3%) | 14 (4%) |
| *Total N in time frame* | *159* | *215* | *374* |

neutered and for 13.5% (N = 33) the neutering status was unknown. Of all female dogs included 43% (N = 55) was intact, 3% (N = 4) was neutered and for 54% (N = 69) neutering status was unknown, with data for each time frame presented in Table 3 below.

In and over both time frames, most dogs were breed classified as a Pitbull type (58%, N = 218; 56%, N = 89 for tf1; 60%, N = 129 for tf2), next as (Belgian/ Dutch/ German) Shepherd type, including Malinois (17%, N = 64; 12%, N = 20; 20%, N = 44), with other dogs typed for instance as Rottweiler, Kangal, Husky, Bernese Mountain Dog.

## Bite incident characteristics

Victims of biting incidents were adults (including police officers or caretakers), children, and/ or animals (other dogs, but also cats, goats, sheep and horses). Most of the incidents preceding confiscation of the dog were with adults or children (also), as shown in Table 4. Where animals were involved, incidents were severe and/ or included a high number of animals killed during one incident.

**Table 3. Descriptive data on sex and neutering status of confiscated dogs.** Dog sex as count and as % of column total for a first time frame (2008–2010), a second time frame (2020-mid-May 2022) and overall for both time frames, for 374 dogs confiscated after a biting incident.

| Dog sex | 2008–2010 | 2020–2022 | Both time frames |
|---|---|---|---|
| | N (% of column total) | N (% of column total) | N (% of column total) |
| Male | 97 (61%) | 148 (69%) | 245 (66%) |
| Female | 61 (39%) | 67 (31%) | 128 (34%) |
| Missing value | 1 | | 1 |
| *Total N in time frame* | *159* | *215* | *374* |
| **Dog neutering status—males** | **2008–2010** | **2020–2022** | **Both time frames** |
| | N (% of column total) | N (% of column total) | N (% of column total) |
| Intact | 72 (74%) | 107 (72%) | 179 (73%) |
| Neutered | 7 (7%) | 26 (18%) | 33 (13.5%) |
| Unknown | 18 (19%) | 15 (10%) | 33 (13.5%) |
| *Total N in time frame* | *97* | *148* | *245* |
| **Dog neutering status—females** | **2008–2010** | **2020–2022** | **Both time frames** |
| | N (% of column total) | N (% of column total) | N (% of column total) |
| Intact | 22 (36%) | 33 (49%) | 55 (43%) |
| Neutered | 1 (2%) | 3 (4%) | 4 (3%) |
| Unknown | 38 (62%) | 31 (46%) | 69 (54%) |
| *Total N in time frame* | *61* | *67* | *128* |

**Table 4. Descriptive data on victims involved in dog biting incidents preceding confiscation of 374 dogs.** Descriptive data on victims of biting incidents involved in biting incidents preceding confiscation of 374 dogs, presented as counts and as % of column total for a first time frame (2008–2010), a second time frame (2020-mid-May 2022) and overall for both time frames.

| Victims | 2008–2010 | 2020–2022 | Both time frames |
|---|---|---|---|
| | N cases (% of column total) | N cases (% of column total) | N (% of column total) |
| Adult or child* | 106 (67%) | 133 (62%) | 239 (64%) |
| *Of which child involved* | *22 (14%)* | *27 (13%)* | *49 (13%)* |
| Dog only | 42 (26%) | 69 (32%) | 111 (30%) |
| Other animal(s), such as horse(s), goats, sheep, cats | 10 (6%) | 12 (6%) | 22 (6%) |
| No information on victim | 1 (1%) | 1 (0%) | 2 (1%) |
| *Total N in time frame* | *159* | *215* | *374* |

* With or without also a dog or another animal victimized

At the biting incident in most cases (63%; N = 235, with no difference between tf1 and tf2) the owner of the dog did not provide assistance to the victim, such as by trying to stop the biting incident, as shown in Table 5.

In most cases (66%, N = 105 for tf1, 77%, N = 166 for tf2) dogs were roaming, including escape from property, at the time of the biting incident. This prevalence was significantly higher in tf2, with 'on leash' applying less to tf2 ($\chi2$ = 16.5, P = 0.001, df = 3). Table 6 presents counts and frequencies for all options of control over the dog, or lack thereof, at the biting incident, and for pairwise comparisons see S1 Appendix in S1 File.

Dog owner reactions upon their dog biting, were only in 14% of cases labelled as cooperative (17%, N = 27 for the first time frame, 12%, N = 26 for the second time frame) as shown in Table 7. Dog owner reactions did not differ significantly between the two time frames (P = 0.21).

For 36% of dogs (42%, N = 66 for the first time frame, 31%, N = 67 for the second time frame) the biting incident leading to confiscation of the dog was the first registered biting incident. For 27% (27%, N = 43 for tf1, 27%, N = 57 for tf2) it was the second biting incident, and the maximal number of previous incidents registered was twelve, see Table 8.

The severeness of dog bites, indicated as based on Ian Dunbar's biting scale and grouped in the categories of moderate, severe or extremely severe biting (see Methods section) did not

**Table 5. Descriptive data on dog owners providing assistance at the biting incident preceding confiscation of 374 dogs.** Descriptive data on dog owners providing ('yes') or not providing ('no') assistance to victim(s) of biting incidents leading to confiscation of 374 dogs, as counts and as % of column total for a first time frame (2008–2010), a second time frame (2020-mid-May 2022) and overall for both time frames.

| Assistance provided by dog owner | 2008–2010 | 2020–2022 | Both time frames |
|---|---|---|---|
| | N (% of column total) | N (% of column total) | N (% of column total) |
| No | 99 (62%) | 136 (63%) | 235 (63%) |
| Yes | 20 (12%) | 32 (15%) | 52 (14%) |
| Yes, but owner left immediately (after dog coming loose of victim) without return or later victim contact | 20 (13%) | 25 (12%) | 45 (12%) |
| No information available | 20 (13%) | 22 (10%) | 42 (11%) |
| *Total N in time frame* | *159* | *215* | *374* |

**Table 6. Descriptive data on dog owners' dog control situation, at biting incidents leading to confiscation of 374 dogs.** Descriptive data on dog owners' dog control situation, at biting incidents leading to confiscation of 374 dogs, presented as counts and as % of column total for a first time frame (2008–2010), a second time frame (2020-mid-May 2022) and overall for both time frames. The dog's roaming applied significantly more so to these situations in the second time frame than the first ($\chi2 = 16.5$, P<0.001, df = 3, and for pairwise comparisons see S1 Appendix in S1 File).

| Control situation | 2008–2010 | 2020–2022 | Both time frames |
|---|---|---|---|
| | N (% of column total) | N (% of column total) | N (% of column total) |
| Roaming dog | **105 (66%)** | **166 (77%)** | 271 (72%) |
| Pulled loose | 11 (7%) | 25 (12%) | 36 (10%) |
| On leash | **34 (21%)** | **19 (9%)** | 53 (14%) |
| No information available | 9 (6%) | 5 (2%) | 14 (4%) |
| *Total N in time frame* | *159* | *215* | *374* |

differ significantly between tf1 and tf2 ($\chi2 = 1.3$, P = 0.51, df = 2 for human-directed bites and $\chi2 = 0.6$, P = 0.74, df = 2 for animal-directed bites) and was more frequently indicated as extremely severe (the highest category) for animal-directed than for human-directed bites, as shown in Fig 1. For human-directed bites overall percentages were 33%, 33%, 37% for moderate, severe, extremely severe biting. For animal-directed bites overall percentages were 10%, 11%, 79% for moderate, severe, extremely severe biting. S2 Appendix in S1 File lists all counts.

## Dog ownership factors associating with biting incidents

For tf1, 426 factors were registered over 159 cases and for tf2 882 factors over 215 cases (total N = 1,308). For the total of 1,308 registered factors, the factors registered most frequently ($\geq$5% of factor total), were those of having a multiple dog household (N = 154), the dog's roaming without the owner (unrelated to biting incident that led to confiscation, N = 106), care task transfer, meaning someone else than the owner walked the dog (N = 98), having a short leash/ muzzle obligation for this dog or a previous dog (N = 72), keeping the dog isolated and/ or in a confined space (N = 66), the dog living with an owner with reported substance abuse (suspected or established; N = 62), owner aggression towards police, bystanders, etc. at confiscation of the dog (N = 60).

**Table 7. Descriptive data on dog owners' reactions upon their dog's biting of a human and/ or animal, at biting incidents leading to confiscation of 374 dogs.** Descriptive data on dog owners' reactions upon their dog's biting of a human and/ or animal, at biting incidents leading to confiscation of 374 dogs, as counts and as % of column total for a first time frame (2008–2010), a second time frame (2020-mid-May 2022) and overall for both time frames. No significant differences were found between the time frames ($\chi2 = 8.5$, P = 0.21, df = 6).

| Owners' reaction | 2008–2010 | 2020–2022 | Both time frames |
|---|---|---|---|
| | N (% of column total) | N (% of column total) | N (% of column total) |
| Laconic | 33 (21%) | 51 (24%) | 84 (22%) |
| Aggressive, threatening and/ or intimidating | 39 (25%) | 37 (17%) | 76 (20%) |
| Not mentioned | 28 (18%) | 47 (22%) | 75 (20%) |
| Cooperative | 27 (17%) | 26 (12%) | 53 (14%) |
| Denial of incident, seriousness or owner role | 21 (13%) | 27 (13%) | 48 (13%) |
| Victim blaming | 10 (6%) | 22 (10%) | 32 (9%) |
| Other such as panic, epileptic seizure or owner being the victim | 1 (0%) | 5 (2%) | 6 (2%) |
| *Total N in time frame* | *159* | *215* | *374* |

**Table 8. Descriptive data on the number of biting incidents reported for dogs in 374 confiscated dogs.** Descriptive data on the number of biting incidents reported for dogs in 374 confiscated dogs, as counts and as % of column total for a first time frame (2008–2010, tf1), a second time frame (2020-mid-May 2022, tf2) and overall for both time frames. Pairwise comparisons between tf1 and tf2 revealed a significantly lower count for dogs confiscated after a first biting incident in tf2 ($\chi2$ = 4.2, P = 0.04, df = 1, residual -2.1). We found a contrasting significantly higher count for dogs confiscated after ≥3 biting incidents in tf2 (42%) than tf1 (32%; $\chi2$ = 4.6, P = 0.03, df = 1, residual 2.1). Note that the incident number includes the last incident that resulted in the dog's confiscation.

| Number of biting incidents reported on for the dog | 2008–2010 | 2020–2022 | Both time frames |
|---|---|---|---|
| | N (% of column total) | N (% of column total) | N (% of column total) |
| One | 66 (42%) | 67 (31%) | 133 (36%) |
| Two | 43 (27%) | 57 (27%) | 100 (27%) |
| Three | 19 (12%) | 39 (18%) | 58 (16%) |
| Four | 17 (11%) | 23 (11%) | 40 (11%) |
| Five to twelve | 14 (9%) | 29 (13%) | 43 (11%) |
| *Total N in time frame* | *159* | *215* | *374* |

When looking at how the factors applied to the cases for tf1, tf2 and overall, we logically see the same factors applying most frequently. For all factors the counts are either unchanged or significantly higher for the second time frame, as shown in Table 9.

For the eight most often seen factors there was a significant difference between the time frames only for the factors of obligatory leashing/ muzzling of the dog and substance abuse by the owner, both indicating an increase in tf2 as compared to tf1, with the other five factors remaining at an unchanged percentage of cases over time.

In line with these findings, multifactorial situations (that is 5 or more factors applying to a case; median, range: 3, 0–16; and see S3 Appendix in S1 File for all distributions) were reported significantly more frequent in tf2 (38%, N = 81 of N = 215) than tf1 (16%, N = 25 of N = 159; $\chi2$ = 21.7, P<0.001, df = 1). When looking at multifactorial situations within our defined four categories (≥2 factors *within a category* applying to a case) of owner household, owner antisocial behaviours, husbandry/ animal view and animal treatment the largest difference between the time frames (all P<0.05, higher prevalence in tf2) was for the category of animal treatment (20% higher, versus 10 or 11% for the other three categories, as shown in Table 10.

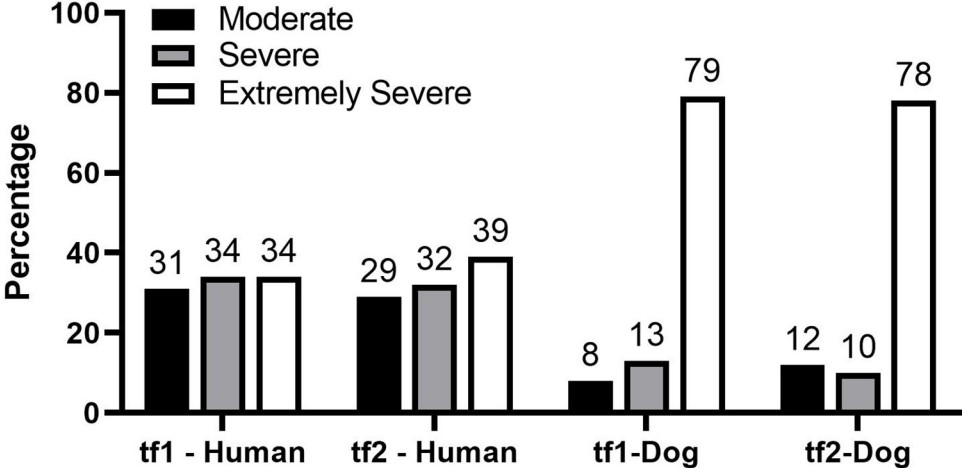

**Fig 1. The severeness of dog bites, indicated based on Ian Dunbar's biting scale for two time frames (2008–2010: tf1 and 2020-2022-mid-May: tf2), with animal-directed more frequently than human-directed bites falling in to the third -most severe- category of biting, that included death as a result.**

**Table 9. Dog ownership factors applying to 374 cases of dogs confiscated for human-/ animal-directed biting incidents.** We present percentages of cases (and counts between brackets) that a dog ownership factor was reported on, sorted from highest to lowest overall frequencies for 30 identified dog ownership factors for a first time frame (2008–2010), a second time frame (2020-mid-May 2022) and overall for both time frames. Pearson Chi-square values are indicated for the pairwise comparisons between the two time frames, per factor (P<0.05, df = 1; significant differences in bold).

| Factor (sorted by most frequent over both time frames) | Category | 2008–2010 | 2020–2022 | All data | Chi-square |
|---|---|---|---|---|---|
| | | % of cases within time frame (N of cases within time frame) | % of cases within time frame (N of cases within time frame) | % of cases overall (N of cases overall) | Pairwise comparisons |
| Multiple dogs in household | Husbandry/ animal view | 40% (63) | 42% (91) | 41% (154) | $\chi2 = 0.3$, P = 0.60 |
| Dog roaming without owner (unrelated to current incident) | Antisocial behaviours | 27% (43) | 29% (63) | 28% (106) | $\chi2 = 0.2$, P = 0.63 |
| Care task transfer | Husbandry/ animal view | 22% (35) | 29% (63) | 26% (98) | $\chi2 = 2.5$, P = 0.11 |
| **Short leash/ muzzle obligation for this/ previous dog** | Antisocial behaviours | 13% (21) | **24%** (51) | 19% (72) | **$\chi2 = 6.5$, P = 0.01** |
| Isolated and/ or confined space | Animal treatment | 14% (22) | 20% (44) | 18% (66) | $\chi2 = 2.8$, P = 0.10 |
| **Substance abuse** | Household | 12% (19) | **21%** (46) | 17% (65) | **$\chi2 = 5.7$, P = 0.02** |
| Animal abuse | Animal treatment | 13% (20) | 20% (42) | 17% (62) | $\chi2 = 3.2$, P = 0.07 |
| Aggression at confiscation of dog | Antisocial behaviours | 16% (25) | 16% (35) | 16% (60) | $\chi2 = 0.2$, P = 0.89 |
| **Antisocial behaviours, e.g. shouting, intimidation** | Antisocial behaviours | 10% (16) | **19%** (41) | 15% (57) | **$\chi2 = 5.7$, P = 0.02** |
| Dog used as weapon or intimidation | Antisocial behaviours | 17% (27) | 11% (24) | 14% (51) | $\chi2 = 2.7$, P = 0.11 |
| Untreated pain and/ or illness signs | Animal treatment | 12% (19) | 13% (28) | 13% (47) | $\chi2 = 0.1$, P = 0.76 |
| Criminal offences | Household | 12% (19) | 13% (27) | 12% (46) | $\chi2 = 0.0$, P = 0.86 |
| **Body Condition Score too low** | Animal treatment | 3% (4) | **17%** (37) | 11% (41) | **$\chi2 = 20.2$, P<0.001** |
| **Owner living situation inadequate** | Household | 4% (7) | **15%** (32) | 10% (39) | **$\chi2 = 10.8$, P<0.001** |
| **Financial issues** | Household | 1% (2) | **15%** (33) | 9% (35) | **$\chi2 = 21.4$, P<0.001** |
| Physical punishment dog | Animal treatment | 6% (10) | 10% (22) | 9% (32) | $\chi2 = 1.8$, P = 0.18 |
| **(Noise) disturbance** | Antisocial behaviours | 3% (5) | **12%** (26) | 8% (31) | **$\chi2 = 9.6$, P = 0.002** |
| Dog 'given' by network | Husbandry/ animal view | 6% (9) | 10% (22) | 8% (31) | $\chi2 = 2.5$, P = 0.11 |
| **Skin/ coat/ nail issues** | Animal treatment | 3% (4) | **13%** (27) | 8% (31) | **$\chi2 = 12.1$, P<0.001** |
| Domestic violence | Household | 9% (14) | 6% (12) | 7% (26) | $\chi2 = 1.5$, P = 0.23 |
| Illogical explanation of biting behaviour | Husbandry/ animal view | 4% (7) | 8% (18) | 7% (25) | $\chi2 = 2.8$, P = 0.10 |
| **Dog's environment unclean** | Animal treatment | 2% (3) | **10%** (21) | 6% (24) | **$\chi2 = 9.5$, P = 0.002** |
| Mental illness | Household | 5% (8) | 7% (14) | 6% (22) | $\chi2 = 0.4$, P = 0.55 |
| Old scars/ white hair markings | Animal treatment | 4% (6) | 6% (13) | 5% (19) | $\chi2 = 1.0$, P = 0.32 |
| **No access to (clean) water/ food** | Animal treatment | 1% (2) | **7%** (15) | 5% (17) | **$\chi2 = 6.9$, P = 0.009** |
| Form of assistance provided to owner | Household | 3% (5) | 4% (9) | 4% (14) | $\chi2 = 0.3$, P = 0.60 |
| **Mentioning of previous dog(s) incidents/ confiscation** | Antisocial behaviours | 1% (1) | **6%** (12) | 3% (13) | **$\chi2 = 6.7$, P = 0.01** |

*(Continued)*

**Table 9.** (*Continued*)

| Factor (sorted by most frequent over both time frames) | Category | 2008–2010 | 2020–2022 | All data | Chi-square |
|---|---|---|---|---|---|
| | | % of cases within time frame (N of cases within time frame) | % of cases within time frame (N of cases within time frame) | % of cases overall (N of cases overall) | Pairwise comparisons |
| Children placed in care, child abuse/ neglect | Household | 4% (7) | 2% (4) | 3% (11) | $\chi^2 = 2.1$, P = 0.15 |
| No access to daylight or covered bench | Animal treatment | 1% (2) | 3% (6) | 2% (8) | $\chi^2 = 1.0$, P = 0.31 |
| Parasites | Animal treatment | 1% (1) | 2% (4) | 1% (5) | $\chi^2 = 1.1$, P = 0.31 |

## Discussion

We studied 374 reports of dogs confiscated for biting a human and/ or dog and found higher or similar prevalence of worrisome dog ownership *factors* in a more recent time frame. A higher prevalence was seen for all four ownership *categories* of household, antisocial behaviours, husbandry/ animal view and animal treatment. When separately assessing the prevalence of the thirty ownership factors, we found nine factors prevalent at a ≥15% rate. Six of these factors were reported at a stable rate over time: having multiple dogs in the household, a dog's roaming, care task transfer -that is the dog being walked by another than the owner-, keeping a dog in an isolated and/ or confined space, suspected or established animal abuse and an owner's general antisocial behaviours in the neighbourhood, such as shouting or name calling. Three of the factors were reported at a higher prevalence in the more recent time frame: an owner's dog being ordered to be short leashed and muzzled in public space, suspected or established substance abuse by the owner and antisocial behaviours such as an owner's shouting at others or intimidating others in public space.

The highest prevalence was for having multiple dogs in the household. Generally, dogs in a multi-animal household are not necessarily provided with lesser 'care', that is management and husbandry levels, as they are reported on in studies on veterinary care [31–33]. However, this characteristic may indicate a mismatch between the dog's care demands and the dog owner's caregiving capacities, including the capacity to prevent a dog's biting. It may also serve as a possible indicator that dog breeding is taking place, as we noticed in several cases in our dataset that young dogs (pups or adolescents) were reported on for multiple dog households. Breeding taking place in a subset of cases where dogs pose serious biting risks is worrying and

**Table 10. Multifactorial situation within four categories of dog ownership factors applying to 374 cases of dogs confiscated for human-/ animal-directed biting incidents.** We present percentages of cases (and counts between brackets) that a dog ownership factors within the categories of owner household, owner antisocial behaviours, husbandry/ animal view and animal treatment was reported on as a multifactorial situation (≥2 factors *within a category* applying to a case) for a first time frame (2008–2010), a second time frame (2020-mid-May 2022) and overall for both time frames. Pearson Chi-square values are indicated for significant differences in pairwise comparisons between the two time frames, per factor (P<0.05, df = 1).

| | | 2008–2010 | 2020–2022 | Both time frames | Chi-square |
|---|---|---|---|---|---|
| | | N (% of column total) | N (% of column total) | N (% of column total) | Pairwise comparisons |
| Household | 0–1 | 88% (140) | 77% (165) | 82% (305) | $\chi^2 = 7.8$, P = 0.005 |
| | ≥ 2 | 12% (19) | **23%** (50) | 18% (69) | |
| Antisocial | 0–1 | 77% (122) | 67% (144) | 71% (266) | $\chi^2 = 4.2$, P = 0.040 |
| behaviours | ≥ 2 | 23% (37) | **33%** (71) | 29% (108) | |
| Husbandry/ | 0–1 | 89% (142) | 79% (169) | 83% (311) | $\chi^2 = 7.5$, P = 0.006 |
| animal view | ≥ 2 | 11% (17) | **21%** (46) | 17% (63) | |
| Animal | 0–1 | 89% (141) | 69% (149) | 78% (290) | $\chi^2 = 19.7$, P<0.001 |
| treatment | ≥ 2 | 11% (18) | **31%** (66) | 22% (84) | |

merits further studying as aggression in dogs has a hereditary component [26, 29], next to the crucial role of for instance the mother dog-offspring bond, socialisation and general early-life experiences [27, 28, 34, 35] as well as the experience of being involved in a biting incidents as a dog, which was seen to associate with dogs becoming 'biting dogs' in the future [8]. The finding of relatively high prevalence of the factors of roaming and care task transfer may also indicate a mismatch between care demands and capacities. Thus, we flag these factors as early warnings of high-risk dog ownership situations.

Two further highly prevalent factors in our dataset seem indicative of animal welfare issues: keeping dogs in an isolated and/ or confined space and suspected or established animal abuse. Dogs need to exercise, for instance through performing foraging behaviours [36–38]. Also, dogs need to perform social interactions [39, 40]. Neither which will be possible for dogs that are kept isolated and/ or confined for prolonged time periods. By not providing the opportunity to the dog to perform species-specific behaviour, the dog's capacity 'to cope and adapt to the demands of the (prevailing) environmental circumstances, enabling it to reach a state that it perceives as positive' will be compromised and welfare might be at stake [41, p.3]. Although we cannot establish if a dog's biting risk is causal or consequential to keeping it confined/ isolated, the relatively high prevalence of the factor in our dataset seems in line with findings in a study on factors associating to dog inflicted fatal bites to humans in the USA. In 76% of these extremely severe biting incidents, biting dogs were kept as resident dogs: dogs kept isolated from regular, positive human interactions [42]. For animal abuse we are similarly unable to establish if a dog's biting risk is causal or consequential to the abuse. However, by the very nature of animal abuse, it will pose a risk to animal welfare. Animal abuse can be defined as the active form of animal cruelty, that is 'any act or omission that contributes to the pain, suffering, or unnatural death of animals, or that otherwise threatens their welfare' [43, p. 355]. Animal abuse can be both physical, for instance through hitting, kicking, strangling, or throwing an animal, as well as psychological [44–48]. Interestingly, next to posing a risk to animal welfare, several studies associate animal abuse to an animal's aggressive behaviours, including biting [42, 44–51] and the high prevalence of animal abuse in our study on severe biting incidents, seems in concord with these earlier findings.

How substance abuse may factor into this relationship between animal abuse and biting incidents seems to warrant further studying. This, as (alcoholic) substance abuse associated to both animal abuse and biting incidents [52–54]. Also, alcohol intake may make potential victims of biting incidents vulnerable, as seen in biting incidents decreasing in three communities in Australia where alcohol restrictions were strengthened [54] and in sixteen victims reportedly being under the influence of alcohol at the time of a deadly dog biting incident [42]. In our study we find substance abuse -alcohol amongst other substances- to be characteristic to a relatively high percentage of the cases (17%), and more so in the more recent (21%) than the more historic (12%) time frame, adding to the argumentation to further study how substance abuse factors into situations of a dog's biting.

Finally, regarding noticeable factors in our study, the often-prevalent factor of an owner's general antisocial behaviours performed in the neighbourhood, in combination with the relatively high prevalence of suspected or established substance abuse, raises questions on the effectiveness of short leash and muzzle orders. An effectiveness of these orders, will largely depend on owners being willing and capable to adhere to the orders, which may be affected by antisocial attitudes or substance abuse. Previous research indicated how deviant behaviours, such as traffic citations and criminal convictions for aggressive crimes, including drugs/ alcohol-related crimes related to opting for a dog at risk of serious biting [55] and confiscated dogs in Sweden, were in 33 of 77 cases confiscated after failure to adhere to an order or ban, such as that of keeping a dog on lead and/or muzzled [56]. Studying the adherence to and effects of

serving orders or bans to dog owners whose dogs present society with severe biting risks, therefore seems highly needed.

It is important to highlight that not all owners of confiscated dogs are unwilling or uncapable to adhere to measures or to follow orders. In a proportion of the here studied cases owners were cooperative and did for instance provide victim assistance when the biting incident took place. These owners could simply be overtaken by a sudden situation where for instance in a recently rehomed dog or in a young dog hereditary traits present themselves strongly for a first time. A possible role for hereditary traits was suggested in an earlier article on confiscated dogs [8]. For these owners, control and educational measures could be foreseen to be effective, but solid scientific evidence on the effect of such educational measures is lacking.

Also, our finding of relatively frequent 'laconic' reactions in owners, may be indicative that in a proportion of cases, an owner's adequate reaction is hindered by psychological processing. In these cases, an owner's psychological processing may be that of a traumatic event hindering an owner's adequate reaction, such as providing victim assistance. Psychological processes seemingly affect dog owners when they are confronted with their dog's biting behaviour, as seen in a study that analysed 484 self-reports on a dog's biting [15]. When dog owners were bitten by their own dog, the inclination to see the incident as 'accidental' or 'unintentional' was more probable than when a person was bitten by an unowned dog [15]. Another recent article explored the role of psychological processes in managing dog aggression risks, such as the emotional state of the owner, social influences and cognitive bias [57]. The authors used protection motivation theory to study how coping strategies were affected by the appraisal of 1) the severity of the threat of dog aggression, 2) the vulnerability to that threat, 3) the effectiveness of threat reduction options and 4) one's own efficacy in deploying threat reduction options [57]. For effective coping strategies dog owners needed to appraise risks as high (probable and severe), as well as judge that risk mitigation strategies were effective and manageable to them. The study regarded less severe aggression and focused on behaviour techniques owners chose in risk mitigation. Nevertheless, the outcome is of interest here, as to point out how psychological factors may affect owner reactions after their dog's biting and how future owner behaviour is (un)likely to change as to mitigate risks. Threat and effectiveness perceptions, as well as barriers to actual deployment of risk mitigation affected how aggressive dog behaviour was managed [57]. This may suggest that for our particular population as well, a dog owner's adequate reaction to prevent and stop biting incidents, could be similarly affected: another interesting point to address in future studies.

## Study limitations

Limitations of the current study are mainly found in its source of study data. We used bite risk assessment reports made for the Dutch national and local government bodies for dogs confiscated after one or more biting incidents (either with humans and/ or animals, including dogs). These reports were made by employees of Utrecht University previously and based on anonymized police or municipality reports on the biting incidents in combination with veterinary reports, made within the first days of a dog's confiscation as to establish any possible medical conditions. This data source implies that reporting bias is probable. The reports used were made by officials, such as police officers and veterinarians, each using official reporting formats to collect and note down findings, which facilitated standardisation and quality. However, particularly between the two time frames, 'zeitgeist', may have affected the original reporting. 'Zeitgeist' in the more recent time frame may have elicited more observation and registration of for instance animal welfare factors. Argumentation is in human-domain professionals being alerted to animal abuse and neglect, as science has pointed out how violence

towards humans and animals may be linked increasingly in past years [58–61]. Also, the field of veterinary forensics seemingly has developed further in recent years, with more interdisciplinary links between this and other forensic disciplines [62–64]. Such increased awareness may increase observation and registration of factors such as a dog's body condition score, skin, coat, nail issues, unclean environment, limited access to (clean) water/ food, which were all seen at higher prevalence in our study's more recent than more historic time frame. Overall, factors regarding animal treatment were reported on more frequently in the recent time frame, as seen in the increase of multifactorial situations being twice as high for the category of animal treatment factors than for the other three categories of factors of household, antisocial behaviours, or husbandry/ animal view, although this category also held most factors per category.

To establish reporting bias, we would need to study our particular population using a different study method and in this light the approach by Patronek et al. [42] to interview the professionals involved in drawing up the original reports, offers an interesting suggestion for such an approach. Finally, we would like to point out that if reporting at higher frequencies in the more recent time frame of factors indeed is based on broader or more diverse reporting of factors, this may be regarded as positive. Broader/ more diverse reporting will allow for enhanced insights on those factors that may be associated to severe biting incidents and can help to ultimately pinpoint those factors to address to prevent and mitigate the risk of severe dog biting incidents.

## Conclusion

In a population of dogs confiscated after human- and animal-directed biting the number of reported worrisome dog ownership factors applying to cases, was higher in recent years. These worrisome dog ownership factors point at societal risks of biting incidents, as well as animal welfare risks—making the involved dogs victims as much as culprits. We indicate that studying effectiveness of an array of measures to prevent biting incidents is urgent, as in recent years a near quarter of cases involved leash/ muzzle obligations without preventing a new biting incident and a fifth of owners reportedly displayed antisocial behaviours unconnected to the dog. We stress that a proportion of involved owners will very likely be willing and capable to adhere to any measure, but that for a sample of owners, a (temporary) unwillingness or incapacity, requires determining which measures for which owners are needed to prevent or cure high-risk situations in dog ownership, with the ultimate aim to make society safer and improve animal welfare.

## Supporting information

**S1 File.**
(DOCX)

**S1 Dataset.**
(XLSX)

## Author Contributions

**Conceptualization:** Ineke R. van Herwijnen, Claudia M. Vinke.

**Data curation:** Ineke R. van Herwijnen.

**Formal analysis:** Ineke R. van Herwijnen.

**Investigation:** Joanne A. M. van der Borg, Chantal M. Kapteijn, Claudia M. Vinke.

**Methodology:** Ineke R. van Herwijnen, Claudia M. Vinke.

**Validation:** Ineke R. van Herwijnen.

**Visualization:** Ineke R. van Herwijnen, Chantal M. Kapteijn.

**Writing – original draft:** Ineke R. van Herwijnen.

**Writing – review & editing:** Ineke R. van Herwijnen, Joanne A. M. van der Borg, Chantal M. Kapteijn, Saskia S. Arndt, Claudia M. Vinke.

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
