## [Decision Letter · Decision Letter 0]

16 Jan 2023

PONE-D-22-30566Factors regarding the dog owner’s household situation, antisocial behaviours, animal views and animal treatment in a population of dogs confiscated after biting humans and/ or other animalsPLOS ONE

Dear Dr. Herwijnen,

Thank you for submitting your manuscript to PLOS ONE. After careful consideration, we feel that it has merit but does not fully meet PLOS ONE’s publication criteria as it currently stands. Therefore, we invite you to submit a revised version of the manuscript that addresses the points raised during the review process.

ACADEMIC EDITOR:

Methods

L177-180: The analysis compares identified factors likely linked to dog bites based on counts of cases between timeframes 2008-2010 and 2020-2022. This determines which factors are consistent or stable across time and which are more prevalent in one timeframe than the other. However, it may be helpful to know which factors across the timeframe will most significantly explain the increase or decrease the frequency of biting incidents. In my opinion, a regression analysis can be done on whether the pre-identified factors (independent variable) influence the number of biting incidents (dependent variable).

Results

L191-194: How about other demographic data that are also relevant, including age, household number, marital status, etc?

Table 9: Write the x^2^ value and p-value instead of n.s.

Other comments: Please include as supplementary a STROBE Statement—a checklist of items that should be included in reports of observational studies (see https://www.strobe-statement.org/checklists/)

We look forward to receiving your revised manuscript.

Kind regards,

Harvie P. Portugaliza, D.V.M., Ph.D.

Academic Editor

PLOS ONE

Journal Requirements:

Reviewers' comments:

Reviewer's Responses to Questions

**Comments to the Author**

1. Is the manuscript technically sound, and do the data support the conclusions?

Reviewer #1: Yes

2. Has the statistical analysis been performed appropriately and rigorously? 

Reviewer #1: Yes

3. Have the authors made all data underlying the findings in their manuscript fully available?

Reviewer #1: Yes

4. Is the manuscript presented in an intelligible fashion and written in standard English?

Reviewer #1: Yes

5. Review Comments to the Author

Reviewer #1: It is a good paper with a very interesting subject to discuss. They presented the rabies from its original and then discussed their results and plans. This paper is attractive and important and the data presented is very interesting.

6. PLOS authors have the option to publish the peer review history of their article (what does this mean?). If published, this will include your full peer review and any attached files.

Reviewer #1: **Yes: **Hashim Talib Hashim

---

## [Author Response · Author response to Decision Letter 0]

8 Feb 2023

Thank you for allowing us to optimize our manuscript ‘Dog ownership factors in a population of dogs confiscated after biting incidents’ for publication by PLOS and thank you to the reviewer for his valuable feedback. Please allow us to address the points raised during the review process, here below. 

Methods:

• L177-180: The analysis compares identified factors likely linked to dog bites based on counts of cases between timeframes 2008-2010 and 2020-2022. This determines which factors are consistent or stable across time and which are more prevalent in one timeframe than the other. However, it may be helpful to know which factors across the timeframe will most significantly explain the increase or decrease the frequency of biting incidents. In my opinion, a regression analysis can be done on whether the pre-identified factors (independent variable) influence the number of biting incidents (dependent variable).

Thank you for this valuable suggestion, which we would like to address in future studies. For the present data set, based on reports on dogs confiscated by Dutch governmental bodies, we feel that any presentation on changes in frequency of biting incidents may unintentionally mislead readers of our manuscript. This as we have investigated a very specific dataset, not reflective of total biting incidents. Also, we feel this manuscript adds value by its reporting on dog ownership factors that possibly underly biting incidents. Our aim is specifically not to add to statistics on biting incidents, but to address which ownership factors may be relevant for future (more hypotheses driven) studies, including studies that could more validly address this interesting suggestion by the reviewer. 

Results:

• L191-194: How about other demographic data that are also relevant, including age, household number, marital status, etc?

Thank you for your suggestion. Unfortunately, this data was not made available in the reports used as the basis of our study.

• Table 9: Write the x2 value and p-value instead of n.s.

Thank you for your suggestion and we have added this information to Table 9. 

Other comments: 

• Please include as supplementary a STROBE Statement—a checklist of items that should be included in reports of observational studies (see https://www.strobe-statement.org/checklists/). 

We have taken another look at the https://www.strobe-statement.org/checklists/ website, but fail to find a checklist that can be filled out for our desk research type study, based on existing risk assessment reports. We are not aware of a STOBE Statement for desk research, but have filled out a cross-sectional STROBE Statement for your convenience. We suggest to not include this as a supplementary file as this may unintentionally lead interested readers into thinking that our study set up was of cross-sectional nature, whilst in fact it regards desk research. 

We have also addressed the additional requirements, regarding style requirements and an extra check of the reference list. With regard to the participant consent, additional details have been added (starting at line 188 in the manuscript with track changes) to the ethics statement. Finally, we have added the minimal data set underlying the results described as a Supporting Information file. 

Hopefully this addresses the raised points satisfactory.

---

## [Editor Report · Decision Letter 1]

21 Feb 2023

Factors regarding the dog owner’s household situation, antisocial behaviours, animal views and animal treatment in a population of dogs confiscated after biting humans and/ or other animals

PONE-D-22-30566R1

Dear Dr. Herwijnen,

We’re pleased to inform you that your manuscript has been judged scientifically suitable for publication and will be formally accepted for publication once it meets all outstanding technical requirements.

Kind regards,

Harvie P. Portugaliza, D.V.M., Ph.D.

Academic Editor

PLOS ONE
---

## [Editor Report · Acceptance letter]

24 Feb 2023

PONE-D-22-30566R1 

Factors regarding the dog owner’s household situation, antisocial behaviours, animal views and animal treatment in a population of dogs confiscated after biting humans and/ or other animals 

Dear Dr. Herwijnen:

I'm pleased to inform you that your manuscript has been deemed suitable for publication in PLOS ONE. Congratulations! Your manuscript is now with our production department. 

Kind regards, 

on behalf of

Dr. Harvie P. Portugaliza 

Academic Editor

PLOS ONE